# Influence of Molasses Residue on Treatment of Cow Manure in an Anaerobic Filter with Perforated Weed Membrane and a Conventional Reactor: Variations of Organic Loading and a Machine Learning Application

**DOI:** 10.3390/membranes13020159

**Published:** 2023-01-27

**Authors:** Khairina Jaman, Syazwani Idrus, Abdul Malek Abdul Wahab, Razif Harun, Nik Norsyahariati Nik Daud, Amimul Ahsan, Shahriar Shams, Md. Alhaz Uddin

**Affiliations:** 1Department of Civil Engineering, Faculty of Engineering, Universiti Putra Malaysia, Serdang 43400, Malaysia; 2School of Mechanical Engineering, College of Engineering, Universiti Teknologi MARA, Shah Alam 40450, Malaysia; 3Department of Chemical and Environmental Engineering, Faculty of Engineering, Universiti Putra Malaysia, Serdang 43400, Malaysia; 4Department of Civil and Environmental Engineering, Islamic University of Technology (IUT), Gazipur 1704, Bangladesh; 5Department of Civil and Construction Engineering, Swinburne University of Technology, Melbourne, VIC 3000, Australia; 6Faculty of Engineering, Universiti Teknologi Brunei, Gadong BE1410, Brunei; 7Department of Civil Engineering, College of Engineering, Jouf University, Sakaka 42421, Saudi Arabia

**Keywords:** sugar content, carbon/nitrogen ratio, biomethane potential test, kinetic models, artificial neural network, electrical energy yield

## Abstract

This study highlighted the influence of molasses residue (MR) on the anaerobic treatment of cow manure (CM) at various organic loading and mixing ratios of these two substrates. Further investigation was conducted on a model-fitting comparison between a kinetic study and an artificial neural network (ANN) using biomethane potential (BMP) test data. A continuous stirred tank reactor (CSTR) and an anaerobic filter with a perforated membrane (AF) were fed with similar substrate at the organic loading rates of (OLR) 1 to OLR 7 g/L/day. Following the inhibition signs at OLR 7 (50:50 mixing ratio), 30:70 and 70:30 ratios were applied. Both the CSTR and the AF with the co-digestion substrate (CM + MR) successfully enhanced the performance, where the CSTR resulted in higher biogas production (29 L/d), SMP (1.24 LCH_4_/gVS_added_), and VS removal (>80%) at the optimum OLR 5 g/L/day. Likewise, the AF showed an increment of 69% for biogas production at OLR 4 g/L/day. The modified Gompertz (MG), logistic (LG), and first order (FO) were the applied kinetic models. Meanwhile, two sets of ANN models were developed, using feedforward back propagation. The FO model provided the best fit with Root Mean Square Error (RMSE) (57.204) and correlation coefficient (R^2^) 0.94035. Moreover, implementing the ANN algorithms resulted in 0.164 and 0.97164 for RMSE and R^2^, respectively. This reveals that the ANN model exhibited higher predictive accuracy, and was proven as a more robust system to control the performance and to function as a precursor in commercial applications as compared to the kinetic models. The highest projection electrical energy produced from the on-farm scale (OFS) for the AF and the CSTR was 101 kWh and 425 kWh, respectively. This investigation indicates the high potential of MR as the most suitable co-substrate in CM treatment for the enhancement of energy production and the betterment of waste management in a large-scale application.

## 1. Introduction

The management of animal waste has become more challenging as production has increased consistently, resulting in an excess of manure that needs to be properly dis-posed of due to growth hormones, antibiotics, heavy metals, and pathogen concentrations that affect both humans and the environment [1,2,3]. Additionally, the inadequate disposal of cow manure (CM) can contribute to the spread of waterborne diseases and pollute the air, land, and water resources [4]. Feedlot cattle can produce manure that is between 5 and 6 percent of their body weight daily, or about 5.5 kg of dry mass per animal; meanwhile, a full-grown milking cow can excrete 7–8% of their total weight in manure each day, or around 7.3 kg of dry mass per animal [5]. In Malaysia, the estimated production of cattle manure was 6.11 million tonnes per year [4]. On the other hand, molasses residue (MR) contains a high organic and sugar content, with 5.3 ± 2.7 and 8.1 ± 2.8% of glucose and fructose, respectively, which makes it a viable substrate for the biogas production [6]. According to Meng et al. [7], 1 tonne of molasses is generated for every 4 tonnes of sugar produced; hence, this substantial wastewater discharge could have disastrous environmental effects if not properly treated. Furthermore, the high organic compound in CM and MR makes these two substrates suitable candidates for use as sources of renewable energy. Nonetheless, greater acidification can happen if there is no adequate system alkalinization, and it can happen in the presence of an uneven substrate availability or organic an overload [6]. Hence, anaerobic digestion (AD) is identified as an option that can provide an efficient and long-term solution for the buildup of these wastes, while also generating biogas (methane) [8,9,10]. CM was frequently utilized as a substrate for the AD system, and it is evident that CM is an efficient feedstock for AD [11,12,13]. Despite the advantage of CM for energy production, its mono-digestion efficiency can be affected by the physicochemical properties of CM, which include a low carbon to nitrogen (C/N) ratio [14] and a high content of indigestible fiber [15], which may result in a poor performance of the digester. To achieve an ideal C/N ratio of 20 to 35 [16,17,18,19], it is therefore highly desirable to have a carbon-rich co-substrate. Additionally, Li et al. [20] found that there were several disadvantages of using CM as a mono-substrate, which include a high lignocellulosic content, a high moisture content, and an abundance of alkaline metals in CM. Meanwhile, sugar-based wastes, which include molasses residue (MR), sweet potato (SP), and sugar beet byproducts (SBP), are hugely produced and often underutilized. MR is regarded as a promising substrate for the generation of biogas among agro-industry byproducts due to its high bioavailability and organic strength [21,22]. It typically contains 45% sugar, which makes it the most appropriate substrate for the generation of biogas [23]. However, the mono-digestion of MR is difficult at high organic loading rates (OLRs) due to the increase in the volatile fatty acid (VFA) accumulation and the nutrient limitation [24] and due to the high salinity in MR [25]. De Vrieze et al. [26] revealed that the bacterial community diminished following the increase in salinity, and a VFA concentration and salinity value higher than 30 mS cm^−1^ resulted in the inhibition of the methanogenesis. Meanwhile, S. Pratt et al. [27] found that the threshold VFA concentration for an AD system was 17 ± 1 gCOD_VFA_L^−1^. Therefore, anaerobic co-digestion (ACoD) has been identified as the option to reduce the inhibition in both substrates and to enhance biogas production [28,29].

Previous research has demonstrated that the ACoD of CM with sugar-based waste boosted the digester’s performance [30,31,32]. Fang et al. [30] investigated biogas production in the AD of desugared molasses co-digested with CM. They reported that the MR co-digested with CM was stable and produced a maximum methane yield. Another study by K. Aboudi et al. [31] reported on the methane generation from the ACoD of sugar beet by-products and CM, with a mix ratio of 50:50; they reported a specific methane production (SMP) rise to 81.4% compared to the mono-digestion of sugar beet by-products. Therefore, it can be implemented to improve biogas production, dilute the inhibitory substance, and decrease the greenhouse gases while keeping the nutritional balance and microorganism synergy [24]. However, there are no previous studies on the investigation of the AD of CM with MR as the co-substrate, and very limited findings on the optimum ratio of these substrates in a continuous study have been reported. Therefore, in this study, CM was used as the main substrate, co-digesting with MR at three different ratios.

In addition, the impact of utilizing various types of reactors and other operational conditions in the AD system was also extensively investigated [33,34]. The anaerobic filter (AF) reactor has been reported to be a little bit more effective and stable than a upflow anaerobic sludge blanket (UASB) reactor. A large number of microorganisms live in the filter bed reactor, as the support material provides sufficient room for a microbial population [35]. Additionally, it has been shown that enhancing the active methanogenic community on the support materials in anaerobic digesters can increase tolerance to inhibitors such as ammonia [25]. Nonetheless, AF, which has a longer startup period and a high washout rate, caused limitation in this system [35,36]. Moreover, the short hydraulic retention time (HRT) causes a limited contact time between the filter media and the substrate, which leads to the reduction in reactor performance [37]. Meanwhile, it was found that the continuous stirred tank reactor (CSTR) is very efficient and is suitable for treating high solid content substrates. It also has a higher organic removal efficiency and produces more methane (CH_4_). Conversely, at high OLR and low HRT, the CSTR’s performance deteriorates [38]. Hence, this research focuses on the continuous study by comparing the AF and the CSTR in treating CM for the mono-digestion and ACoD of CM, MR.

In addition to the experimental work, modelling, which includes kinetic studies and artificial neural networks (ANNs), of the AD of CM and MR was also investigated and analyzed. Designing commercial-scale digesters can benefit from modelling since it provides a useful idea from which to initiate the design and operation and can be used to anticipate the AD system behavior and maximize the biogas output [39]. Kinetic studies use the application of pre-determined formulas [40,41]; meanwhile, the ANN is an intelligent system which involves computation and mathematics, inspired by the biological nervous system (human brain processes), and produces the equation (output) for the biogas prediction [42,43]. In addition, several studies have applied kinetic studies to the ACoD of CM with sugar-based products. Gómez-Quiroga [44] applied the modified Gompertz model for the ACoD of sugar beet pulp and cow manure. Ohuchi et al. [45] demonstrated the anaerobic ACoD of sugar beet tops silage with cow manure followed by kinetic studies using a first order model. These two studies demonstrated only one kinetic model in their investigation. There were no previous studies comparing the best-fit kinetic model of the ACoD of CM a with sugar-based substrate. Meanwhile, Mougari et al. [42] applied two models, an ANN and a modified Gompertz model, to compare the accuracy between the kinetic study and the ANN for various organic waste. They found that both approaches were proven to perform well in estimating biogas and methane production, with the ANN having a smaller deviation and a slightly better result with a satisfactory R^2^ value of 0.9998 and an RMSE of 0.0047. Despite the advantages of the kinetic model and the ANN, none of the previous work has compared and analyzed these algorithms in the AD of CM and MR.

Aiming at the further investigation of the performance of the mono-digestion and ACoD of CM and MR and the implementation of machine learning as a forecasting tool, the objectives of the study were therefore: (1) to evaluate the performance of the mono-digestion of CM and the ACoD of CM with MR; (2) to compare the performance of CSTR and AF on the AD of CM and MR; (3) to identify the optimum ratio of the ACoD of CM and MR; (4) to identify the best-fit kinetic model and to develop a well-established ANN model to forecast biogas production; (5) to compare the accuracy of these two machine learning applications from the point of view of R^2^ and RMSE; and (6) to predict electrical energy generation in an on-farm application.

## 2. Materials and Methods

### 2.1. Physiochemical Characterization of CM, MR, and CM + MR

Table 1 depicts the characteristics of the substrate used for the biomethane potential (BMP) test and the continuous study of CM, MR, and the co-digestion of CM + MR; meanwhile, Table 2 shows the total sugars in the CM and MR. For the BMP test, apart from the mono-digestion study of CM and MR, the ACoD of these substrates was conducted with a mixing ratio of 50:50, as recommended by Aboudi et al. [31], who provided the optimal ratio of 50:50 for the CM and sugar-based product.

The feeding regime for the continuous system involves four different ratios, including 100:0, 50:50, 30:70, and 70:30 for CM and MR to identify the optimum mixing ratio for ACoD in a continuous system using the CSTR and the AF. The ratio measured in the BMP test and the continuous study was derived based on the percentage of volatile solid. 

A cow farm, ‘Ladang 16′ at Universiti Putra Malaysia, Serdang, Selangor was the location for the CM sample collection. The farm does have a separation system for both urine and manure. Meanwhile, the MR sample was supplied by Forward Energy Sdn Bhd, a private company located in Ipoh, Perak, Malaysia. The MR sample was collected in the final stage of the molasses wastewater treatment. To prevent the samples from deteriorating at room temperature, they were kept at 4 °C. The samples were then examined for the carbon to nitrogen (C/N) ratio, total nitrogen, volatile solids (VS), total solids (TS), total dissolved solids (TDS), chemical oxygen demand (COD), oil and grease (O&G), color, and salinity.

### 2.2. Biomethane Potential (BMP) Test without Filter (WoF) and with Filter (WF)

Two alternative operating conditions, including a system without filter (WoF) and another with filter (WF), were used for the BMP test. These operating conditions were adopted to mimic the CSTR (without filter) and the AF (with filter). Both operating conditions were run under mesophilic temperatures which were maintained by placing the BMP bottles in a water bath. The temperature was continuously monitored by an automated temperature digital panel and a manual thermometer.

For the BMP WoF, nine 1 L BMP bottles were employed. These bottles included CM exclusively (Bottles 1, 2, 3), MR solely (Bottles 4, 5, 6), and co-digest CM + MR (Bottles 7, 8, 9). All BMP bottles were filled with pure nitrogen gas (N_2_) and then sealed using rubber stoppers to remove any remaining air from the head space. The bottles were then incubated in a water bath at 38 °C. The same setup was adopted for the BMP WF, which was fed with CM (Bottles 10, 11, 12), MR (Bottles 13, 14, 15), and CM + MR (Bottles 16, 17, 18). The setup for the BMP bottles is shown in Table 3. The support carrier in the BMP WF was fabricated with synthetic grass to act as a filter media for the microbes. The synthetic grass used was 25 mm synthetic grass (SG) with a perforated weed membrane (BLS HOME DECO, Perak, Malaysia) 25 mm 1 m × 1 m, cut into circular shape, to fit the BMP bottles at the bottom. For the BMP bottles with ACoD, the BMP WoF (digester 3), and the BMP WF (digester 6), the mixing ratio used was 50:50.

The volume of biogas generated was measured using a graduated measuring cylinder and the water displacement method as reported by Dhamodharan et al. [46]. In addition, two control BMP bottles containing inoculum and filter medium were constructed as a control for this experiment. The volatile suspended solid (VSS) content was 116.54 mg/L at the Universiti Putra Malaysia sewage treatment facility, where the inoculum was obtained. The substrate to inoculum (S/I) ratio for both WoF and WF was fixed at 1:4, since a low S/I ratio is favorable for methanogenesis [47]. The setup for BMP WoF and BMP WF is illustrated in Figure 1a,b. The configuration of the synthetic grass and perforated weed membrane in the AF reactor is depicted in Figure 1c,d. The material of the synthetic grass is polypropylene with a water or flow permeability of 4.7 × 10^−3^ cm/s. The opening size and tensile strength are 202 μm and 10.6 kN/m, respectively.

### 2.3. Continuous Study 

Two different anaerobic reactors were operated for a continuous study, as displayed in Figure 2 and Figure 3. Both reactors were acclimatized with synthetic wastewater for 2 weeks before feeding with the actual sample. The OLR was kept low during acclimatization (0.5 g/L/day). The VS was monitored until a stable value was obtained; then, the feeding with the actual substrate could be carried out. Table 4 summarizes the experimental design and the ACoD ratio. 

#### 2.3.1. Continuous Stirred Tank Reactor (CSTR) Setup

The CSTR consists of borosilicate glass as the digester’s tank, with a height of 465 mm and an inner diameter of 650 mm, as shown in Figure 2; the working volume of the CSTR is 5 L. It is connected to the water displacement method through the gas outflow tube at the top of the reactor. To keep the reactor warm at 38 °C, the CSTR was covered with a silicone rubber heater, with an attached thermostat to maintain the temperature. The reactor was also equipped with a stirrer for the intermittent stirring of the substrate and inoculum. Additionally, an inlet at the top for the feeding port and an outlet at the bottom for the digested collection was also equipped. The hydraulic retention time (HRT) was adjusted in accordance with the OLR. It was seeded with inoculum, as previously described in Section 2.2.

#### 2.3.2. Anaerobic Filter (AF) Setup

Figure 3 depicts the setup for the AF. The reactor, with a working capacity of 12 L, was built using PVC tubes with a cylinder shape; the width was 214 mm and the height was 740 mm. Similarly to the CSTR, the biogas produced from the AF reactor was measured using the water displacement method, which was connected by the gas outflow tube. The reactor was equipped with an effluent discharge outlet at the top and an influent inlet at the bottom. Surrounding the sludge was the synthetic grass with a perforated weed membrane (SG) (BLS HOME DECO, Perak, Malaysia), with a thickness of 25 mm; it was 1 m × 1 m at the bottom and top and had a round cylindrical form at 200 mm from the bottom. The purpose of utilizing the SG was to provide a medium for microbial growth. Additionally, a top flat and round perforated PVC was attached to the SG to improve settling and lessen sludge washout. A rubber silicone heater was used to keep the temperature at 38 °C, and a thermostat was attached to the silicone heater to monitor the temperature. The reactor was run in an up-flow mode. A peristaltic pump was employed as a means of providing a feeding mechanism for the reactor, where the substrate was pumped from the bottom through the sludge blanket at a constant rate of 10 rpm so that there was enough time for contact between the substrate and the inoculum [48]. Depending on the applied OLR, the HRT was varied. The reactor was inoculated with inoculum, as previously described in Section 2.2.

### 2.4. Analytical Method

Daily pH measurements were performed on the digestate from each reactor. In addition, the total alkalinity (IA/PA) ratio, the total ammonia nitrogen (TAN), and the volatile solids (VS) removal were measured twice each week. The amount of biogas was measured everyday using the water displacement method described in Section 2.3. The TDS, pH, and salinity were measured using a Meter tabletop with a stirrer from ISOLAB, Eschau, Germany. Meanwhile, the VS and VSS were calculated utilizing the standard method for the examination of water and wastewater [49] and in accordance with the protocol outlined in [36,50]. The salicylate powder pillow technique 8155 was implemented to measure the TAN using a spectrophotometer (HACH 157 DR 900, Agilent, Santa Clara, CA, USA). In the meantime, 0.02 N sulfuric acid (H_2_SO_4_) was employed with the titrimetric technique to assess the IA/PA ratio. A gas chromatograph (Agilent HP 6890 N, Santa Clara, CA, USA) fitted with a thermal conductivity detector (TCD) and a capillary column (30 mm × 0.5 mm × 40 m) was utilized to quantify the biogas generated composition. The VFAs were measured using a Shi-madzu GC 2010 gas chromatograph, Kyoto, Japan. The machine is equipped with a flame ionization detector (FID) and a FameWax capillary column (30 m × 0.32 mm × 0.25 µm). 

### 2.5. Kinetic Analysis

The kinetic study was analyzed using Microsoft Office 2021 Excel Solver, where all the kinetic parameters can be predicted for all the models (modified Gompertz model (MG), logistic model (LG), and first order model (FO)). The peak SMP rate and the lag phase can be derived by using the MG and LG; meanwhile, the FO can explain the hydrolysis stage [39]. In a batch study, the methane generation rate was estimated by the MG model to be proportional to the population growth rate of the methanogens. The LG, however, believed that the methane production was proportional to its maximum rate [39]. On the other hand, the FO model was used to describe hydrolysis as the rate limiting phase in the AD system [51]. The formulas used are shown in Table 5.

where

*R_m_* = maximum biogas generation rate (L/d);λ = lag phase (day); *S*(*t*) = cumulative biogas production at digestion time “*t*” days;*S* = biogas potential of the substrate (L); *K* = biogas production rate constant;*t* = time (days).

In order to determine the substrate’s capability to produce biogas, an Excel solver model was used. The model fit for the experimental data was represented by the root mean square error (RMSE), which was computed using Equation (4). A low RMSE value indicates a better fit
(4)$$RMSE=\sqrt{\frac{\sum_{i=1}^{N}{\left(S_{exp,i}-S_{mod,i}\right)}^{2}}{N}}\;$$
where *S_exp_*_,*i*_ is the biogas production (average) from the experiment; *S_mod_*_,*i*_ is the biogas production (model); and *N* is the total number of data [52].

Finally, the correlation coefficient (*R*^2^) value was determined, which indicated the accuracy of the actual line to the predicted line. The actual and forecasted data have an acceptable correlation when the *R*^2^ value is close to 1.

### 2.6. Artificial Neural Network (ANN) Analysis

The ANN was developed to forecast the production of biogas; it is a type of machine learning system that uses variables to create mathematical relationships between sets of defined input and output data without the need for prior physical knowledge. It is used in a variety of applications to address challenging problems, including forecasting, modelling, cluster analysis, trend recognition, simulation, and others [42]. Three layers make up the ANN structure: an input layer that collects data, an output layer that produces calculated values, and hidden layers that connect the input and output layers [43]. A neuron, which serves as the foundational processing component of an ANN, collects inputs and generates output; this is illustrated in Figure 4.

The sigmoid activation function (log-sigmoid or tan-sigmoid) from Equations (5) and (6) is multiplied with the outcomes. The values generated by one of these two sigmoid functions are sent to the output layer; they have the same number of neurons and output variables and are controlled by a linear activation function (pure linear), as defined by Equation (7) [42].
(5)$$g\left(a\right)=Tanh\left(a\right)=\frac{exp\;\left(a\right)-exp\;\left(-a\right)\;}{exp\;\left(a\right)+exp\;\left(-a\right)\;}$$
(6)$$g\left(a\right)=log\left(a\right)=\frac{1}{1+e^{-a}}$$
(7)g(a)=Purelin(a)=a

In this study, the ANN was applied to estimate the daily biogas production using the following parameters, including OLR, VS_in_, the C/N ratio, C, and N for the 18 different BMP bottles (3 replicates of six sets of BMP bottles). The architecture of the network is indicated in Table 6, where it features two hidden layers: the first utilizes the tan-sigmoid activation function on its 31 neurons, while the second layer employs the log-sigmoid activation function on its 20 neurons. MATLAB software is used to train the network from the dataset. The input dataset was chosen during the training phase and should fall between −1 and 1 [43]. The input data were normalized using Equation (8).
(8)$$X_{nor}=\frac{3\left(X_{i}-X_{min}\right)}{2\left(X_{max}-X_{min}\right)}-1$$
where *X_i_* is original value; *X_min_* and *X_max_* are the minimum and the maximum value of the original values, and *X_nor_* is the normalized value.

### 2.7. Projection of Electrical Energy

The projection of electrical energy was computed from the SMP recorded at different OLRs. The SMP (methane yield in L CH_4_/gVS_added_) was converted into m^3^/kgVS and kJ/kgVS, as reported by Khanal et al. [53]. Therefore, the projected electrical energy generation (kWh) was calculated using Equation (11):(9)$$\mathrm{Energy}\;\mathrm{in}\;E\;\left(\frac{\mathrm{kJ}}{\mathrm{kgVS}}\right)=\mathrm{SMP}\;\left(\frac{m^{3}}{\mathrm{kgVS}}\right)\times 35,846\;\left(\frac{\mathrm{kJ}}{m^{3}}\right)$$
(10)$$\mathrm{Energy}\;\mathrm{in}\;P\left(\frac{\mathrm{kWh}}{\mathrm{kgVS}}\right)=E\;\left(\frac{\mathrm{kJ}}{\mathrm{kgVS}}\right)\times 0.00028\;\left(\frac{\mathrm{kWh}}{\mathrm{kJ}}\right)$$
(11)$${\mathrm{Energy}\;\mathrm{in}\;P}_{e}\;\left(\mathrm{kWh}\right)=P\left(\frac{\mathrm{kWh}}{\mathrm{kgVS}}\right)\times \mathrm{VS}\;\mathrm{added}\;\left(\mathrm{kgVS}\right)$$
where VS_added_ is the volatile solids of the substrate (kgVS per day).

## 3. Results and Discussion

### 3.1. Summary of BMP Test of AD for CM, MR, and CM + MR

The single substrates of CM and MR and the co-digestion of CM + MR BMP tests were conducted to evaluate the possibility of the generation of biogas in two different operational conditions. Table 7 displays the results of the BMP test for the samples WoF and WF. The IA/PA ratios for all the batch digesters were below the threshold value of 0.3, with the range value of 0.13 to 0.25, which shows that the system was stable. The pH value shows a similar trend, where it is within the suitable range of 6.8 to 8.5 [31], with the highest value of 8.21 obtained by MR WF and the lowest of 6.75 by CM WoF. For TAN, all the digesters were also below the threshold of 1500 mg/L, with the highest value of just 295 mg/L by CM + MR WF. Meanwhile, the highest SMP and VS removal was recorded in the BMP digesting the MR WF with an SMP of 45.05 mL CH_4_/VS and a VS removal of 67.7%, respectively. By comparing the operational conditions of WoF and WF, it can be observed that using a filter in the digester increased the VS removal in all the BMP tests. Similarly, the SMP shows a higher value in the system equipped with a filter for all of the substrates. Hence, it can be concluded that introducing filter media elevated the performance of the low solids sample. Likewise, the previous researchers also agreed that the digestion of liquid substrate showed a better performance in a system with filter media. This is caused by the abundance of methanogens, favoring biomass acclimation and improving acetic acid consumption [54,55,56].

### 3.2. Continuous Study of CM, MR, and CM + MR at increasing OLR

#### 3.2.1. Biogas Production and pH at Different OLRs

CM is one of the important bioresources for AD, and some efficient strategies have been applied to increase the AD efficiency, such as the ACoD of CM with sugar-based products [30,31,32]. The CSTR and AF reactor continuously operated for 106 days, and an increment of OLR was conducted after the biogas production was stable. In this study, the mono-digestion of CM was run in the CSTR and the AF reactor at OLR 1 gVS/L/day for 12 days. Throughout the process, the OLR was increased gradually until it was 4 gVS/L/day for both CSTR and AF. Then, the substrate was added with MR at a ratio of 50:50 (CM:MR), from OLR 4 g/L/day to OLR 7 g/L/day (CSTR) and OLR 6 g/L/day (AF).

Overall, the AD system showed a relatively stable and good performance for the biogas production and pH value. As depicted in Figure 5, the biogas increased gradually in the mono-digestion system in the CSTR, which produced biogas of 13 L at OLR 4 g/L/day. Consequently, MR was added to the substrate (CM) at the same OLR (50:50 ratio), and the system had a positive effect, with an increment of biogas of 84%. This finding indicates the limitation of CM as a single substrate in AD due to the imbalanced C/N ratio and the higher content of easily degradable organics in MR compared to CM [25,30]. These findings are in line with the C/N in CM reported in this study, which is 9.01. Comparing the sugar composition of CM and MR, it can be observed that galactose, mannose, and fructose are present in CM, while mannose, fructose, and xylose are in MR. Hence, the presence of xylose in MR may contribute to higher biogas production during co-digestion. Wyman et al. [57] mentioned that xylose is an easily fermentable compound due to its simple structure; thus, there was the increase in biogas production when MR was introduced. The system remained stable until OLR 6 g/L/day at the same ratio of ACoD. Conversely, the reactor was clearly inhibited after OLR 7 g/L/day, where the biogas production dropped from 29 L to 18 L. The inhibition at a high OLR may be due to salinity stress; Chen et al. [25] reported that methanogenesis is inhibited when the salinity is above 30 mS/cm, which is proven by the salinity of the MR obtained in this study (34 mS/cm).

Likewise, the pH also decreased gradually from 7.26 to 6.41, which was the result of the VFA accumulation [58]. This was supported by the VFA obtained in Table 8 and Figure 6 for OLR 7 g/L/day at a ratio 50:50, where the highest values of the VFAs were recorded: acetic acid (328.982 mg/L), butyric acid (55.554 mg/L), and propionic acid (371.419 mg/L). Conversely, at the optimum OLR (OLR 5 g/L/day), the acetic acid and propionic acid were recorded as 9-fold and 3-fold less than in OLR 7 g/L/day, respectively. From the point of view of the mixing ratios, the total VFA accumulation significantly reduced by 400 mg/L, which indicates a stable system at a ratio 70:30 compared to 50:50. Moreover, the drop in pH value could also be attributed to the long-chain fatty acids (LCFAs) piled up in the system [59]. The sudden decrease in pH in the system causes the inhibition of the syntrophic bacteria, acetogen and methanogen, hence the unstable system [60]. The inhibition at OLR 7 g/L/day can be explained by the excess amount of substrate due to a high loading rate, causing insufficient time for hydrolysis in the CSTR reactor [22]. This could also be due to the high sodium and potassium concentration in MR [61], the unsuitable C/N ratio of the system, etc., which showed a negative effect on the biogas production.

The pH has a significant impact on the rate at which microorganisms convert organic matter into biogas. According to Morales-Polo et al., anaerobic populations flourished at pH 4 to 8.5, while methanogenic bacteria thrived at pH 6.5 to 7 [62]. Meanwhile, Egwu et al. [63] found that maintaining the pH range between 7.0 and 7.2 in the CSTR enhanced the activities of the methanogenic bacteria. In this study, the pH of the digestate was within 6.0–8.0, and no buffer or alkaline materials were added during the AD process.

To overcome the signs of inhibition at OLR 7 g/L/day, the mixing ratio of the substrate was changed to 30:70 (CM:MR). At this phase, the average daily biogas production was 15 L and pH 6.10. Further inhibition was observed at this ratio. For this reason, a mixing ratio of 70:30 (CM:MR) was introduced. At this phase, the reactor exhibited the significant improvement of biogas production up to 100% (13 L to 27 L), as well as the improvement of the pH. This indicates that among the three ratios of the ACoD of these two substrates, the best performance was achieved with a 70:30 ratio. Hence, it can be seen that the ratio of 70:30 is superior to the 50:50 and 30:70 ratios. However, it was found that at the ratio of 50:50, a better SMP and biogas production was obtained. This shows that the 70:30 ratio is suitable for treating waste in large amounts at a stable rate, while the 50:50 ratio is better for producing methane gas (SMP), even at a small OLR.

In a similar manner to the CSTR, the AF also demonstrates the same trend, where increasing the OLR causes an increment in biogas production. However, the system shows inhibition at OLR 6 g/L/day, where the biogas production drops sharply, and the pH value also increases. The AF also shows improvement when co-digested with MR. During this phase, biogas was produced at a steady phase, with a gradual increment from OLR 1 to OLR 6. Co-digesting with MR at the same OLR (OLR 4) with a 50:50 (CM:MR) ratio caused the biogas production to increase with a difference of 69%. Comparing the CSTR and the AF, it can be observed that the CSTR performed better. In a previous section (BMP), it was found that implementing a filter resulted in a better performance. However, in a continuous system, the opposite trend was observed. This could be caused by the fact that in BMP there is sufficient time for the microbes to attach to the filter media. However, in AF the feeding was conducted every day; so, it was fairly difficult for the biomass to be retained on the filter media. This theory is proven by a study by Daud et al. [37], who explained that short HRTs led to a washout of biomass or sludge in a UASB reactor. This results in a short period of contact between both the sludge bed and the substrate, which causes the biological process in the reactor to break down. In addition, the pH value was recorded to be very low (6.22) towards the end of the investigation, which was not suitable for methanogens to grow since the optimum pH range is between 6.8 and 7.8 [64].

#### 3.2.2. VS removal and Specific Methane Production (SMP) at Different OLRs

As depicted in Figure 7, the average of the VS removal for the CSTR was above 50% throughout the operation. At OLR 2, the VS removal achieved 85%, which was the highest, and slightly reduced to 70% and remained stable up to OLR 6. The VS removal dropped dramatically and recorded the lowest value of 32.61% at OLR 7 (50:50 CM:MR). At a mixing ratio of 70:30, the VS removal significantly increased to 69%. Additionally, comparing between mono-digestion and ACoD, the results indicated that ACoD with the same OLR (OLR 4) was preferable when compared to mono-digestion. This is proven with the higher value of the VS removal in ACoD at OLR 4 (88.5%) compared to the mono-digestion (79.5%).

The VS removal in the AF was recorded at an average of 70% in the mono-digestion phase. A further evaluation of ACoD was conducted at OLR 4 with a mixing ratio 50:50. This investigation revealed that the AF displays very poor VS removal when co-digested with MR, reaching values below 20%. The low VS removal could be attributed to the acidification and hydrolysis that produced VFA and the ethanol that was kept in the effluents [65]. The trend of VS removal is proportional to the methane production for both the CSTR and the AF.

Comparing the AF and the CSTR, the SMP was recorded as being higher in the CSTR compared to the AF. At the initial OLR in the mono-digestion phase, the SMP in the AF increased gradually, peaking at 0.53 LCH_4_/VS_added_ at OLR 4 (ACoD), but after ACoD, the SMP stayed constant without any increment or decrement. On the other hand, when ACoD was added at OLR 4, the CSTR displayed a larger SMP value (1.19 LCH_4_/VS_added_). The improvement of the SMP values is supported by the previous studies [66,67,68,69]. The highest SMP recorded in the CSTR was 1.24 LCH_4_/VS_added_ at OLR 5 g/L/day. Hence, this shows that the optimum OLR for the CSTR is OLR 5 (ACoD), and for the AF, it is OLR 4 (ACoD). The CSTR reactor is clearly more appealing than the AF reactor for actual application due to its higher biogas generation, VS removal, SMP, and stable pH.

#### 3.2.3. IA/PA Ratio and Total Ammonia Nitrogen at Different OLRs

Figure 8 depicts the TAN concentrations for both the CSTR and the AF, which elevate with the increasing of the OLR. Both digesters recorded the highest value of 310 mg/L (OLR 5) for the CSTR and 164 mg/L (OLR 2) for the AF. According to earlier research, the inhibitory thresholds of TAN are between 1500 and 2500 mg/L [69,70]. Hence, both reactors remained within the threshold range. A low TAN value at higher OLRs indicates that the protein degradation was not taking place, as a result of the low amounts of acetogens and methanogens [36]. Meanwhile, the higher TAN in the CSTR indicates the quick degradation of the substrate by the abundant microbial biomass [36]. As for the IA/PA ratio, both reactors displayed a steady value that was generally below the 0.3 threshold. However, as the OLR grew, the IA/PA ratio also rose dramatically, peaking at 2.01 and 1.1 for the CSTR and the AF, respectively, both of which were at OLR 7.

According to Fuentes et al. [71], the pH value alone is insufficient to interpret and analyze the stability resulting from the wastewaters’ buffer capacity. The IA/PA ratio is thus an appropriate measure for this purpose. The IA/PA ratio below 0.3 is advised to maintain a stable system [72]. For the CSTR, Figure 8 demonstrates that starting on day 67 the process exhibited a decline in the reactor stability, with the IA/PA rising and the TAN falling rapidly at OLR 7, with an ACoD ratio of 50:50. This suggests an imbalance in the AD process. On the other hand, starting on day 39 (ACoD), the AF began to exhibit a process imbalance, following a similar pattern in the CSTR. The CSTR reactor was more effective than the AF at converting the organic substrate (CM and MR) to methane at a higher OLR, as evidenced by the higher TAN reading throughout the process. During this phase (OLR 7 with ACoD ratio of 70:30), the IA/PA ratio dropped to 0.18, which is within the optimum range, and the TAN increased to 110, which indicates an increase in the methanogenic activity.

### 3.3. Kinetic Analysis of Biogas Production from BMP Test

The MG, LG, and FO models were applied to extract the kinetic parameters for all the substrates; this is presented in Table 9 and Figure 9. The rate constant (k) for the biogas generation was computed using FO, while MG and LG were both utilized to obtain Rm. Overall, the FO model, which had the lowest RMSE and the highest R^2^ when compared to the other models, provided the best-fit model to the experimental data for both samples. This finding is in line with several past research studies [73,74]. The FO models’ reaction rates (k) have a wide range of values, ranging from 0.002 to 0.2 d-1. For CM, the first order model performs the best for both WoF and WF. This is owing to the smallest RMSE values of 12.451 and 16.592 for CM WoF and CM WF, respectively. Additionally, The R^2^ value was consistently above 0.9 and very close to 1, which further supports the notion that there is little discrepancy between the experimental and kinetic values. The MG model and the LG model are the least suitable models for CM WoF and CM WF, respectively, because they produced the greatest RMSE values of 47.658 for CM WoF and 187.128 for CM WF.

For MR mono-digestion, the first order model fits MR WF the best, while the LG model fits MR WoF the best, which is attributed to the fact that the RMSE calculated was the lowest (43.597 for WoF and 24.216 for WF). The MG model and the LG model for the MR WF and the MR WoF substrates, respectively, show the least applicability since they recorded the greatest RMSE values of 415.908 for MR WoF and 227.667 for MR WF. Nevertheless, the first order model for both CM + MR WoF and CM + MR WF best described the experimental data for the ACoD of CM with MR. The RMSE values of the samples are 74.343 and 75.483, respectively, and the R^2^ value is high, at a remarkable 0.9413 and 0.9382, respectively. A study by Gómez-Quiroga [44] found that when using the modified Gompertz model the R^2^ value obtained was 0.9852 when sugar beet pulp was mono-digested and 0.9888 when co-digested with CM at a 50:50 ratio. In another study, Ohuchi et al. [45] obtained an R^2^ of 0.9938 for CM alone and 0.9880 and 0.9862 for the ACoD of sugar beet silage at a ratio of CM to sugar beet of 60:40 and 40:60, respectively.

### 3.4. Artificial Neural Network (ANN) Analysis for BMP Test

Two ANN networks were developed for the prediction of biogas production in BMP. ANN2 exhibits better R^2^ and MSE values (0.97164, 0.00027) compared to ANN1 (0.94, 0.0069), as depicted in Figure 10 and Figure 11. ANN1 was developed using the equation combination of the tan-sigmoidal and the log-sigmoidal for the first and second hidden layers, respectively. Meanwhile, a log-sigmoidal and tan-sigmoidal equation combination was applied for ANN2.

Another investigation was carried out to draw a comparison between the results of the ANN and kinetic models based on their predictive capability and fitness accuracy. The performance indicates the RMSE, measured with Equation (4) and R^2^, which were defined with excel tools and were used to measure the predictive accuracy of the models. Figure 12 depicts the experimental and predicted biogas production using the best ANN model. It is observed that the predicted biogas production shows a good accuracy with the experimental value. The values of MSE, RMSE, and R^2^ for the ANN model achieved 0.00027, 0.0164, and 0.97164, which is shown in Figure 11. When compared to the kinetic analysis, the ANN shows better performance in predicting the biogas production for the BMP study. This is because the ANN obtained a higher R^2^ value and a lower RMSE value of 0.97164 and 0.0164, compared to the highest average R^2^ and the smallest average RMSE obtained by the first order model, which were 0.94035 and 57.204, respectively. This finding is supported by the previous studies, which compare the performances of kinetic models and ANNs [42,75].

### 3.5. Electrical Energy Generated from Laboratory Scale (LS) and on-Farm Scale (OFS) Projection

Energy generation was assessed to investigate the ability of the substrate using two types of reactors for electrical energy generation. The result obtained is summarized in Table 10 and proved that the application of ACoD enhanced energy generation, generating 0.0818 kWh and 0.0422 kWh for the CSTR and the AF, respectively, at OLR 4. Both the CSTR and the AF produced higher energy as the OLR increased; however, the CSTR decreased at OLR 6 and continued to deteriorate at OLR 7. The highest electrical energy produced by the CSTR was at the optimum OLR (OLR 5) and was 0.1065 kWh. Comparing the three different ratios at OLR 7, the highest energy generated was 0.0637 kWh at a ratio of 70:30, which proves that it was the best ratio for CM and MR. For the estimation of the energy generation potential in the on-farm scale (OFS) system, the proposed design was considered to be 20 m^3^ in volume. The working volumes for the lab scale (LS) in this study were 0.012 m^3^ (AF) and 0.005 m^3^ (CSTR). Hence, using the same ratio of substrate to working volume, the volume of the substrate used in the OFS was computed, and the electrical energy generation was calculated as shown in Table 10. In comparison with a previous study on the co-digestion of chicken dung and food waste [39], the optimum OLRs for both the CSTR (OLR 5) and the AF (OLR 4) produced higher electrical energy generation, whereas the previous study achieved 0.002 kWh [39].

## 4. Conclusions

The results from this study showed that in both the CSTR and the AF, the ACoD of CM with MR performs better compared to mono-digestion at OLR 4 g/L/day, with increment of 84% and 76% for biogas production and SMP, respectively, in the CSTR. Likewise, AF shows an incremental cost of 69% for biogas production. Both the CSTR and the AF result in a stable TAN and IA/PA ratio of 110 mg/L and 0.284 for the CSTR and 64 mg/L and 0.154 for the AF. This clearly shows a stable operating condition that is adequate to prevent the IA/PA (alpha) from rising above 3. Therefore, the presence of xylose in MR and a suitable C/N in the ACoD of CM and MR enhanced the digester stability and tolerated a high OLR. Additionally, the CSTR successfully achieved a better performance than the AF reactor, where it maintained a more stable and higher biogas production with SMP (1.01 LCH_4_/gVS_added_) until OLR 6 g/L/day, whereas the AF achieved a 60% lower biogas production and SMP at the same OLR. A further load rise to OLR 7 demonstrated signs of inhibition at the ACoD ratio of 50:50 (CM:MR). It was observed that the comparison of the three different ratios revealed that the optimal mixing ratio was 70:30 (CM:MR). For biogas forecasting, FO was identified as the best-fit kinetic model and ANN1 (tan-sigmoid and log-sigmoid combination) demonstrated better regression. The comparison between the kinetic models and the ANN proved that the ANN is better at predicting biogas production. This is attributed to the higher R^2^ value and the smaller RMSE values of 0.97164 and 0.164, compared to highest average R^2^ and the smallest average RMSE obtained by the FO model, which were 0.94035 and 57.204, respectively. The highest electrical energy recorded in the CSTR was at the optimum OLR 5 g/L/day, based on the highest SMP (0.1065 kWh) in the ACoD, which was higher than the mono-digestion (0.0351 kWh). Meanwhile, the AF recorded the highest electrical energy of 0.0398 kWh at OLR 2 g/L/day in the mono-digestion phase, 90% higher than the ACoD, which indicates that the waste availability for the electrical energy generation is an additional advantage. The OFS energy generation potential revealed that 425 kWh can be recovered if the ACoD system is employed. Additionally, the optimization of biogas and methane production in a large-scale system can be implemented through the dynamic application of the FO models, the ANN, and the projected electrical energy from the OFS system.

## Figures and Tables

**Figure 1 membranes-13-00159-f001:**
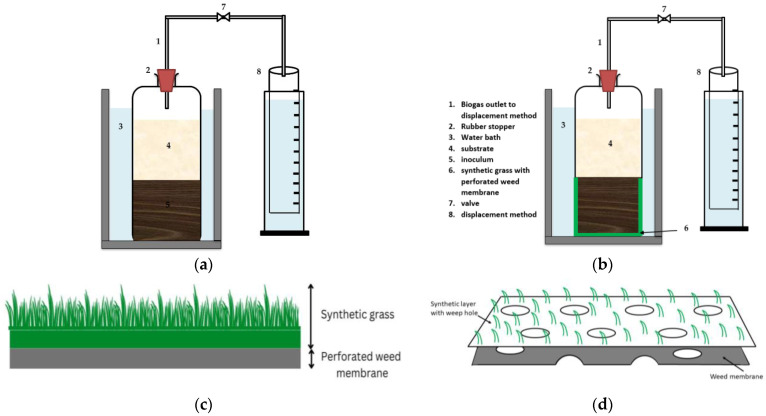
(**a**) Schematic diagram of BMP WoF; (**b**) schematic diagram of BMP WF; (**c**) cross-section of synthetic grass; (**d**) details of perforated weed membrane.

**Figure 2 membranes-13-00159-f002:**
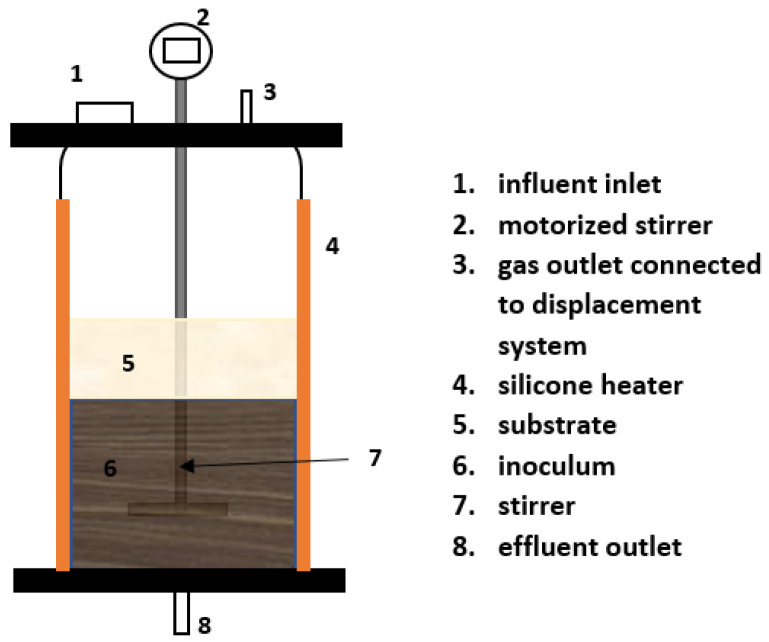
Schematic diagram of continuous stirred tank reactor (CSTR).

**Figure 3 membranes-13-00159-f003:**
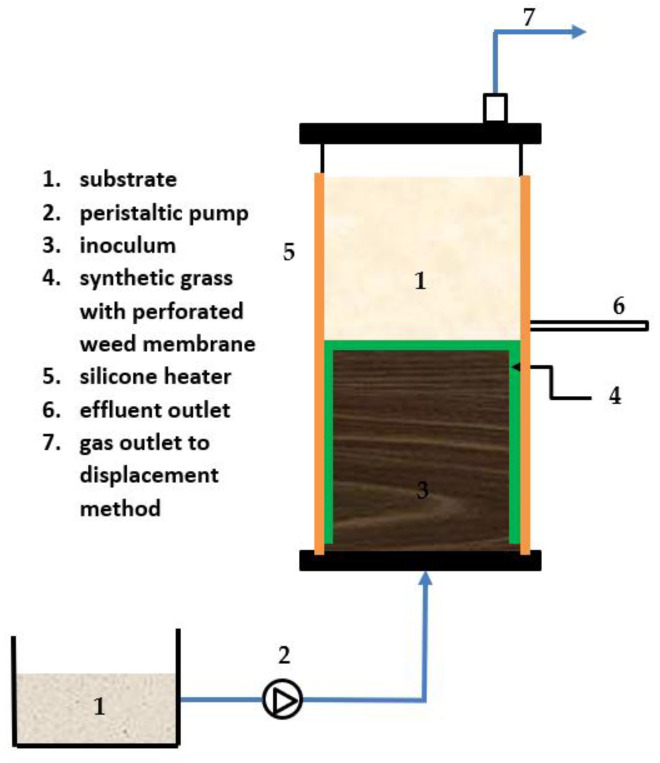
Schematic diagram of anaerobic filter with perforated membrane (AF) reactor.

**Figure 4 membranes-13-00159-f004:**
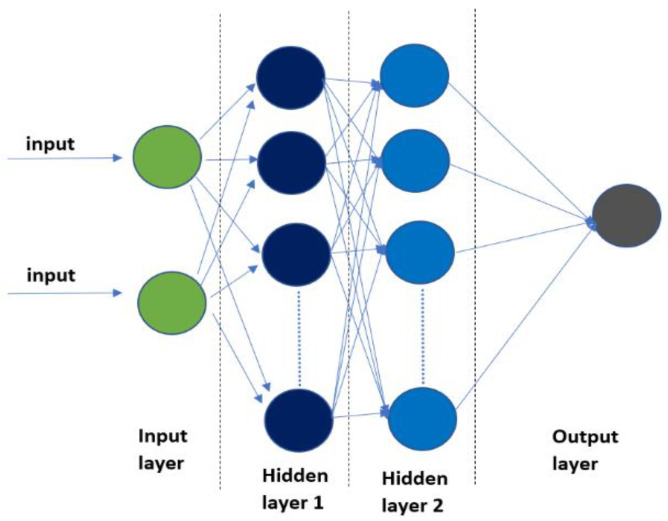
Architecture of artificial neural network (ANN).

**Figure 5 membranes-13-00159-f005:**
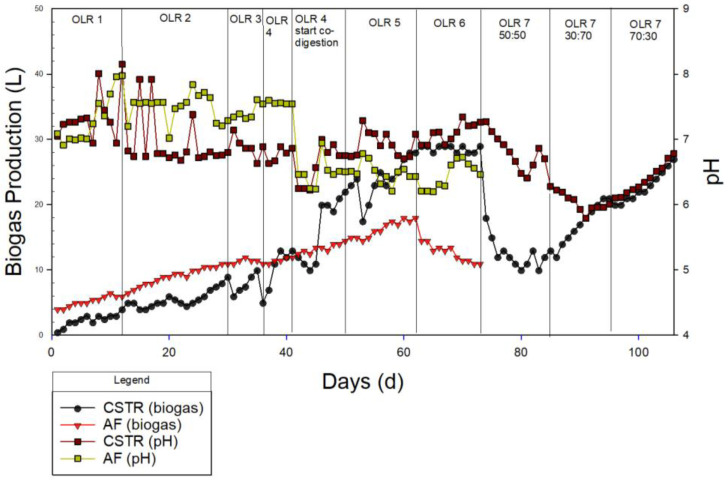
Biogas production and pH of continuous system.

**Figure 6 membranes-13-00159-f006:**
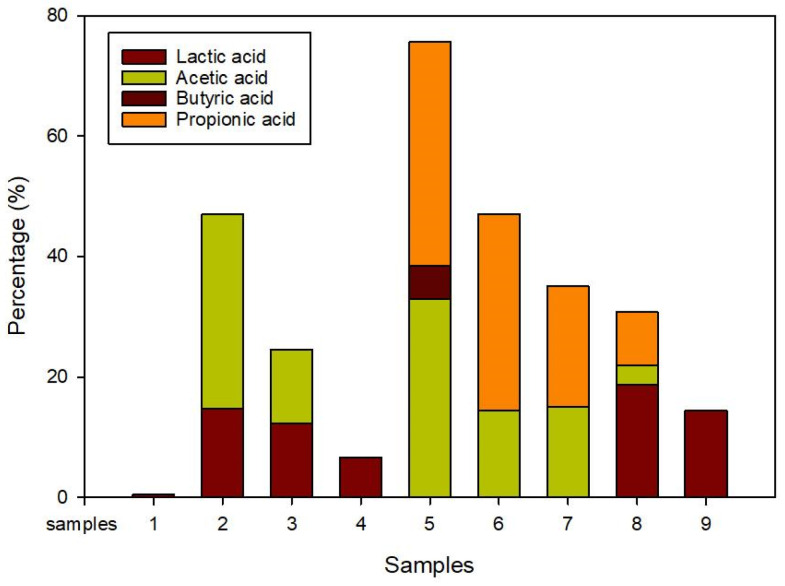
The proportion of 4 volatile fatty acids (VFAs) during different organic loading rates (OLRs).

**Figure 7 membranes-13-00159-f007:**
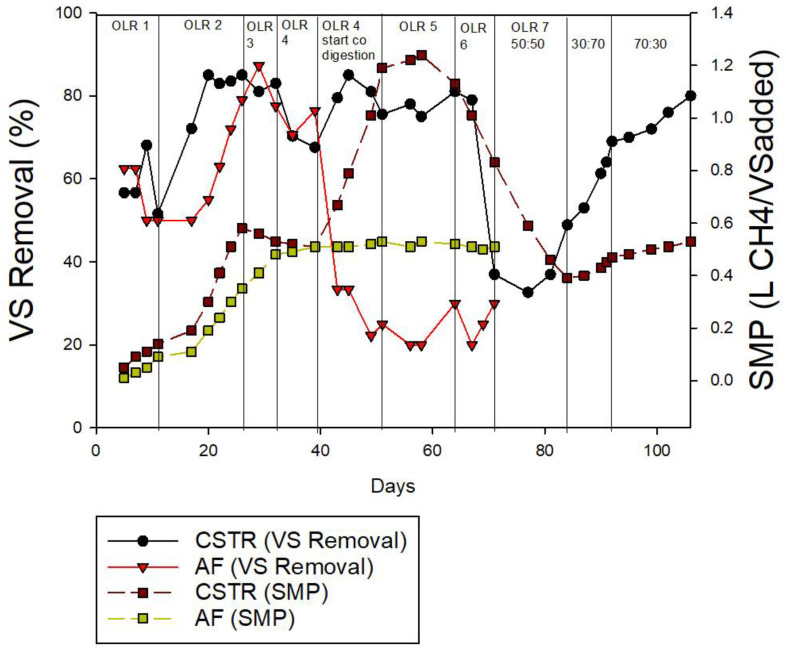
Volatile solids (VS) removal and specific methane production (SMP) for continuous study.

**Figure 8 membranes-13-00159-f008:**
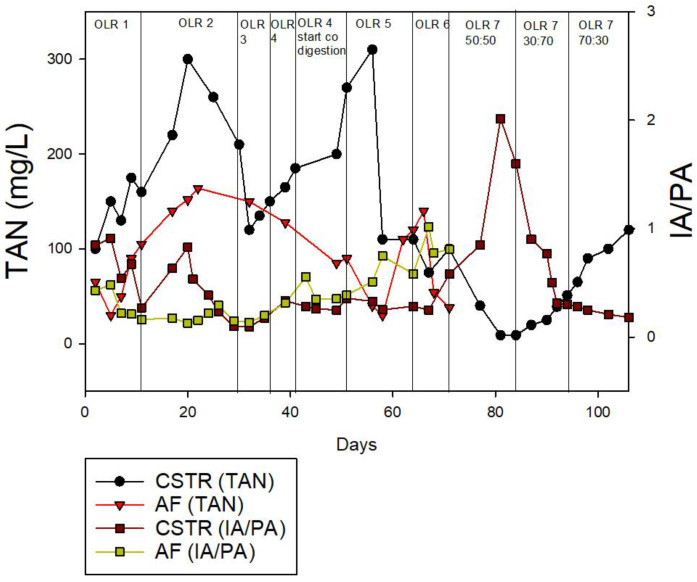
Total ammonia nitrogen (TAN) and total alkalinity ratio (IA/PA) for continuous study.

**Figure 9 membranes-13-00159-f009:**
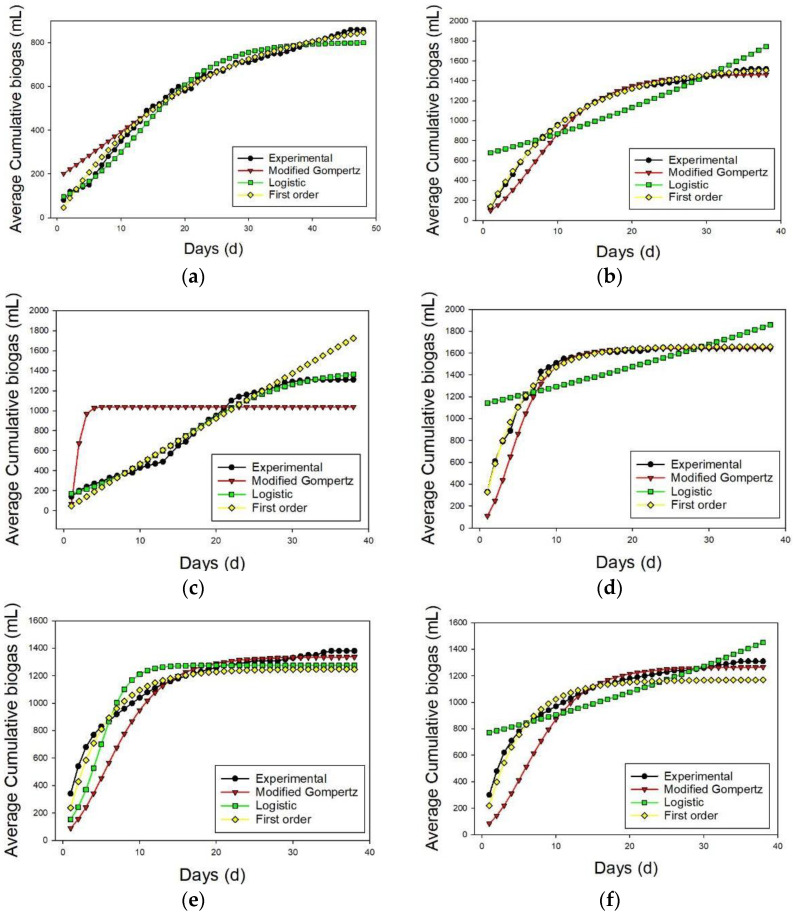
Kinetic study for CM, MR, and CM + MR (average from 18 bottles): (**a**) CM WoF, (**b**) CM F, (**c**) MR WoF, (**d**) MR F, (**e**) CM + MR WoF, (**f**) CM + MR F.

**Figure 10 membranes-13-00159-f010:**
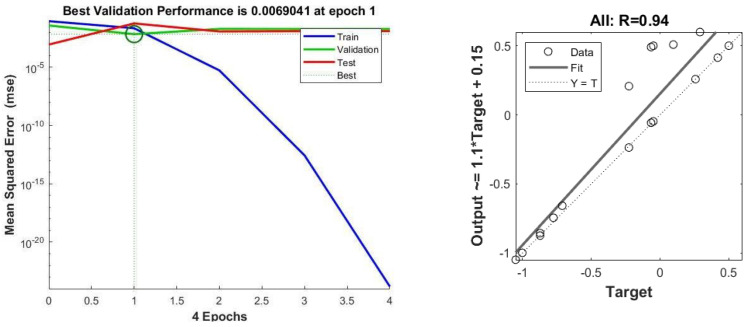
MSE and R^2^ value for ANN1.

**Figure 11 membranes-13-00159-f011:**
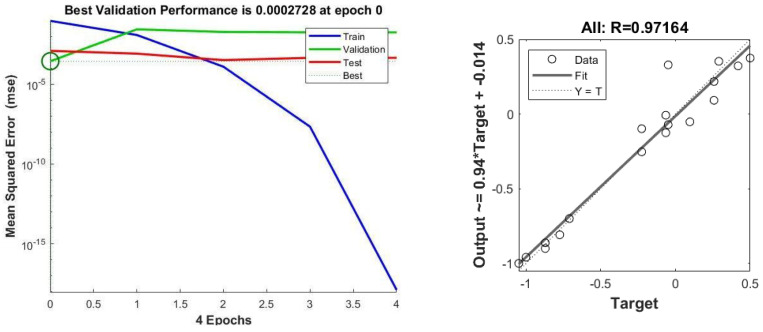
MSE and R^2^ value for ANN2.

**Figure 12 membranes-13-00159-f012:**
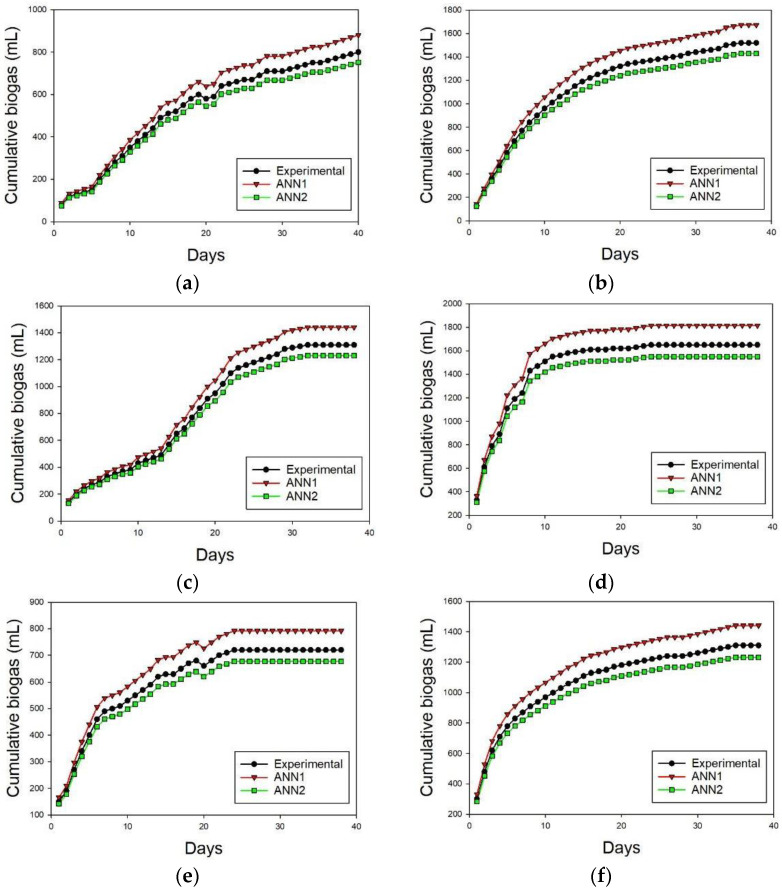
ANN graph fitting for CM, MR, and CM + MR: (**a**) CM WoF, (**b**) CM WF, (**c**) MR WoF, (**d**) MR WF, (**e**) CM + MR WoF, (**f**) CM + MR WF.

**Table 1 membranes-13-00159-t001:** Physicochemical characteristics of raw cow manure (CM), molasses residue (MR), and co-digestion of cow manure and molasses residue (CM + MR).

Parameters	Unit	CM	MR	CM + MR (50:50)	CM + MR (70:30)	CM + MR (30:70)
COD	mg/L	128,000 ± 16,000	70,400 ± 18,400	99,200 ± 17,000	110,720 ± 16,720	87,680 ± 17,680
TN	mg/L	14,200 ± 1300	2500 ± 180	8350 ± 220	10,690 ± 870	4435 ± 240
VS	mg/L	399,590 ± 33,440	44,645 ± 12,770	444,235 ± 32,790	293,106 ± 27,230	151,128 ± 18,970
TS	mg/L	470,360 ± 33,690	46,250 ± 13,770	457,290 ± 23,730	329,252 ± 27,700	173,480 ± 19,740
TDS	mg/L	7000 ± 940	22,925 ± 2100	14,962 ± 1370	55,870 ± 7210	37,050 ± 4290
O&G	mg/L	42,300 ± 7000	1402 ± 200	21,220 ± 1200	30,030 ± 4900	13,670 ± 2240
Color	Pt-Co	81,000 ± 11,000	42,750 ± 6500	61,875 ± 4300	69,525 ± 9650	54,225 ± 7850
Salinity	mS/cm	11.6 ± 1.6	33.75 ± 12.00	22.67 ± 2.00	18.24 ± 2.68	27.10 ± 4.12
C/N ratio	-	9.01	28.16	11.88	10.36	19.78

**Table 2 membranes-13-00159-t002:** Total Sugars in raw CM and MR.

Parameters	Unit	CM	MR
Sucrose	mg/L	-	-
Glucose	mg/L	-	-
Galactose	mg/L	0.292	-
Mannose	mg/L	0.013	0.007
Fructose	mg/L	0.010	0.004
Xylose	mg/L	-	0.213

**Table 3 membranes-13-00159-t003:** Biomethane Potential (BMP) test setup.

BMP Bottle	Operating Condition	Substrate
1, 2, 3	WoF	CM
4, 5, 6	WoF	MR
7, 8, 9	WoF	CM + MR
10, 11, 12	WF	CM
13, 14, 15	WF	MR
16, 17, 18	WF	CM + MR

**Table 4 membranes-13-00159-t004:** Experimental design and operating condition of continuous study.

Reactor	Period (Days)	OLR (g/L/Day)	ACoD Ratio
CSTR	12	1	100% CM
	18	2	100% CM
	6	3	100% CM
	6	4	100% CM
	10	4	50% CM/50% MR
	11	5	50% CM/50% MR
	11	6	50% CM/50% MR
	11	7	50% CM/50% MR
	11	7	30% CM/70% MR
	11	7	70% CM/30% MR
AF	12	1	100% CM
	18	2	100% CM
	6	3	100% CM
	6	4	100% CM
	10	4	50% CM/50% MR
	11	5	50% CM/50% MR
	11	6	50% CM/50% MR

**Table 5 membranes-13-00159-t005:** Kinetic Model for anaerobic digestion (AD).

Model	Mathematical Definition	Source	Equation
First Order	S (t)=S [ 1−exp (−kt)}	[40,41]	(1)
Modified Gompertz Model	$S\left(t\right)=Sexp\;\left\{-exp\left[\frac{R_{m}}{S}exp\left(\lambda -t\right)+1\right]\right\}$	[40]	(2)
Logistic Model	$S\left(t\right)=\frac{S}{1+exp\left[4Rm\;\left(\frac{\lambda -t}{S}\right)+2\right]}$	[41]	(3)

**Table 6 membranes-13-00159-t006:** Detailed information about the developed artificial neural network (ANN) models.

Parameters	ANN1	ANN2
No. of layers	2	2
No. of neurons first hidden layer	20	20
No. of neurons second hidden layer	20	20
Activation function first hidden layer	Tan-sigmoid	Log-sigmoid
Activation function second hidden layer	Log-sigmoid	Tan-sigmoid

**Table 7 membranes-13-00159-t007:** BMP Performance.

	CM		MR		CM + MR (50:50)	
	WoF	WF	WoF	WF	WoF	WF
IA/PA ratio	0.147 ± 0.002	0.228 ± 0.005	0.155 ± 0.006	0.139 ± 0.010	0.250 ± 0.008	0.192 ± 0.011
pH	6.75 ± 0.05	6.93 ± 0.15	7.76 ± 0.11	8.21 ± 0.08	7.10 ± 0.05	7.26 ± 0.09
TAN (mg/L)	168 ± 20	255 ± 60	235 ± 40	198 ± 55	170 ± 30	295 ± 65
SMP (mL CH_4_/VS)	25.90 ± 8.20	35.28 ± 5.80	33.65 ± 3.30	45.05 ± 7.40	32.92 ± 6.20	31.24 ± 5.30
VS removal (%)	38.5 ± 5.50	49.6 ± 2.80	42.3 ± 3.60	67.7 ± 2.40	51.4 ± 1.80	64.5 ± 0.91
Volume of biogas (mL)	860 ± 80	1520 ± 120	1310 ± 180	1650 ± 60	720 ± 100	1310 ± 200

**Table 8 membranes-13-00159-t008:** Volatile Fatty Acids (VFA) during different organic loading rates (OLRs) for continuous stirred tank reactor (CSTR) and anaerobic filter with perforated membrane (AF) reactor.

Substrates	Unit	OLR 4 (Mono)		OLR 4 (Co)		OLR 7 (Co)			Optimum OLR	
		CSTR	AF	CSTR	AF	CSTR			CSTR	AF
		100:0	100:0	50:50	50:50	50:50	30:70	70:30	OLR 5 (Co) (50:50)	OLR 2 (Mono) (100:0)
Lactic acid	mg/L	5.684	147.826	122.885	66.7	-	-	-	187.653	144.328
Acetic acid	mg/L	-	352.459	122.885	-	328.982	144.672	150.954	31.471	-
Butyric acid	mg/L	-	-	-	-	55.554	-	-	-	-
Propionic acid	mg/L	-	-	-	-	371.419	326.082	199.539	88.789	-

**Table 9 membranes-13-00159-t009:** Summary of kinetic analysis using different models.

Model	Parameter	Units	Sample					
			CM WoF	CM WF	MR WoF	MR WF	CM + MR WoF	CM + MR WF
Modified Gompertz	R-square		0.9853	0.9857	0.1398	0.9896	0.9379	0.679
	RMSE		47.658	74.363	415.908	110.942	76.554	152.183
	Rm	L/day	21.763	98.327	700.001	217.266	57.70	102.599
	Mean R-square							0.83937
	Mean RMSE							146.268
Logistic	R-square		0.9792	0.7580	0.9895	0.4859	0.8608	0.742
	RMSE		32.849	187.128	43.597	227.667	120.517	122.868
	Rm	L/day	33.121	45.556	50.030	37.195	175.455	33.175
	Mean R-square							0.80257
	Mean RMSE							122.438
First Order	R-square		0.9932	0.8239	0.9512	0.9943	0.9413	0.9382
	RMSE		12.451	16.596	140.132	24.216	74.343	75.483
	Rate Constant (k)	d^−1^	0.0507	0.0958	0.002	0.219	0.210	0.208
	Mean R-square							0.94035
	Mean RMSE							57.204

**Table 10 membranes-13-00159-t010:** Summary of electrical energy yield for samples of CM and CM + MR in different OLRs.

Sample	OLR (g/L/Day)	SMP (LCH_4_/gVS_added_)		Energy (KWh/kgVS_added_)		Electrical Energy Generation (kWh) [LS]		Electrical Energy Generation (kWh) [OFS]	
		CSTR	AF	CSTR	AF	CSTR	AF	CSTR	AF
CM	1	0.14	0.09	1.39	0.90	0.0024	0.0018	9.45	3.00
	2	0.58	0.35	5.78	3.49	0.0199	0.0140	78.61	23.27
	3	0.53	0.48	5.28	4.78	0.0273	0.0287	105.6	47.80
	4	0.51	0.51	5.08	5.08	0.0351	0.0203	140.21	67.73
CM + MR	4 (50:50)	1.19	0.53	11.85	5.28	0.0818	0.0422	145.73	70.40
	5 (50:50)	1.24	0.51	12.35	5.08	0.1065	0.0508	425.83	84.67
	6 (50:50)	1.01	0.51	10.06	5.08	0.1040	0.0610	416.08	101.6
	7 (50:50)	0.39		3.88	-	0.0468	-	187.32	-
	7 (30:70)	0.47		4.68	-	0.0565	-	224.64	-
	7 (70:30)	0.53		5.28	-	0.0637	-	253.44	-

## Data Availability

The data presented in this study are available on request from the corresponding author.

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
