# Peer review of "Influence of Molasses Residue on Treatment of Cow Manure in an Anaerobic Filter with Perforated Weed Membrane and a Conventional Reactor: Variations of Organic Loading and a Machine Learning Application"

_membranes, 2023, doi:10.3390/membranes13020159_

Round 1

Reviewer 1 Report

The Comments

Manuscript Number:. Membranes-2169378

Title: Influence of sugar-rich molasses residue on treatment of cow manure in an Anaerobic Filter reactor with Perforated Weed Membrane and a Conventional Reactor: From start-up to optimum organic loading rate and a machine learning application

Reviewer

General Comments

This review aims to propose Influence of sugar-rich molasses residue on treatment of cow manure in an Anaerobic Filter reactor with Perforated Weed Membrane and a Conventional Reactor: From start-up to optimum organic loading rate and a machine learning application. The manuscript is proper to achieve the target of the study. The review survey is well done, but they do not address some important points in the manuscript as stated below.

Specific Comments

·       Modify the title of the manuscript as this title is too long with not clarify the main objective of the manuscript.

·       Reduce the numbers and words of the keywords with trying to put some key words not related with the title.

·       The first statement of the abstract is very twisted, you should be precise.

·       The language needs to be modified as the writing is very poor and the sentences too long and very twisted.

·       Line 14: why bet-ween, it is one word and the same in line 18, 39, 112, 133, .

·       Line 46: what is the relation of united states with your manuscript, you should mention the problem in your country (Malaysia)

·       Line 62: (To achieve an 62 ideal C/N ratio of 20 to 35 [17], it is therefore highly desirable to have a carbon-rich co- 63 substrate). This statement has been mentioned by many authors not only one such as (https://doi.org/10.1016/j.biombioe.2020.105947)

·       Line 95: Anaerobic biofilter has been recently used by researches, mention some of his advantages and some studies working on anaerobic biofilter for treatment of waste such as (10.21608/EJCHEM.2021.67579.3493)

·       Table 2: what is the reason that CM contain sugars more than MR?

·       Figure 5: Why you did not complete with AF reactor, this comparison is unfair; you should stop the experiment at day 70, if you want to do fair comparison? as the biogas of CSTR declined and become more and more after the adaptation of the bacteria why you stop the experiment of AF ? Clarify.

·       The same question for figure 7 & 8.7

·       Regarding the models, why you did it for only 50 days however, your work was around 100 days.

·       Remove figure 13, you do not need it.

·       The article needs a thorough review of the English language.

Author Response

General Comments

This review aims to propose Influence of sugar-rich molasses residue on treatment of cow manure in an Anaerobic Filter reactor with Perforated Weed Membrane and a Conventional Reactor: From start-up to optimum organic loading rate and a machine learning application. The manuscript is proper to achieve the target of the study. The review survey is well done, but they do not address some important points in the manuscript as stated below.

Thank you very much sir. The authors appreciate your kind comments. We have taken note that, and corrections were affected accordingly throughout the manuscript. Thank you once again for the great suggestions.

  1. Modify the title of the manuscript as this title is too long with not clarify the main objective of the manuscript.

Response 1: Many thanks for your great suggestion, Sir. We have rephrased the title and reduced the number of words from 37 to 30.

  1. Reduce the numbers and words of the keywords with trying to put some key words not related with the title.

Response 2: Very good observation Sir. We have reduced the keywords and included new keywords.

  1. The first statement of the abstract is very twisted, you should be precise.

Response 3: Noted Sir, your suggestion is highly appreciated.  We have touched up the first statement of the abstract.

  1. The language needs to be modified as the writing is very poor and the sentences too long and very twisted.

Response 4: Many thanks for your great suggestion, Sir. We have rephrased the sentences throughout the manuscript.

  1. Line 14: why bet-ween, it is one word and the same in line 18, 39, 112, 133, .

Response 5: Very good observation sir. We have corrected the words.

  1. Line 46: what is the relation of united states with your manuscript, you should mention the problem in your country (Malaysia)

Response 6: We could not agree more, Sir, we have replaced the US data to Malaysian data.

  1. Line 62: (To achieve a 62 ideal C/N ratio of 20 to 35 [17], it is therefore highly desirable to have a carbon-rich co- 63 substrate).This statement has been mentioned by many authors not only one such as (https://doi.org/10.1016/j.biombioe.2020.105947)

Response 7: Noted sir, we have included 3 new citations to support this statement, which include:

  1. https://doi.org/10.1016/j.biombioe.2020.105947
  2. https://doi.org/10.1016/j.biortech.2019.02.122
  3. https://doi.org/10.1016/J.FUEL.2018.07.094

8. Line 95: Anaerobic biofilter has been recently used by researches, mention some of his advantages and some studies working on anaerobic biofilter for treatment of waste such as (21608/EJCHEM.2021.67579.3493)

Response 8: We couldn’t agree more with you sir. We have added the advantages and disadvantages of anaerobic filter in line 91 until line 99. Many previous studies have used anaerobic filter with various types of filter media including plastic and bed rock media. Nevertheless, this is the only study which utilize synthetic grass with perforated membrane as filter media.

  1. Table 2: what is the reason that CM contain sugars more than MR?

Response 9: Thank you for your observation. We are aware that the total sugar concentration is slightly lower in MR compared to CM (0.091mg/L difference). Furthermore, the molasses residue that we have used in this study is the final stage of wastewater produced from molasses production. We have also added this information in the methodology section in line 162. Nevertheless, we would like to emphasize the presence of xylose in MR has further added the sugar composition in CM. For your information, we have removed the word ‘sugar-rich’ in the title and throughout the manuscript. The explanation of the role of xylose in AD is shown in line 380-385.

  1. Figure 5: Why you did not complete with AF reactor, this comparison is unfair; you should stop the experiment at day 70, if you want to do fair comparison? as the biogas of CSTR declined and become more and more after the adaptation of the bacteria why you stop the experiment of AF? Clarify.

Response 10: Thank you for the question, As observed in figure 5, AF performance inhibit at OLR 6, meanwhile CSTR still performed well, and that was the reason we stopped the observation for AF and continued increasing the OLR for CSTR. The AF reactor shows little improvement after co-digestion but further increment of OLR has caused sign of inhibition. This is proven by result in TAN (average 120 mg/L) (low methanogenic activity) and IA/PA ratio (more than 1) (low buffering capacity ).

  1. The same question for figure 7 & 8.7

Response 11: Thank you, sir, for this question, previous response is referred. Additionally, VS removal and SMP (figure 7) also shows similar pattern, where low SMP (0.51 LCH4/gVSadded) and VS removal (30%) is obtained by AF at OLR 6.

  1. Regarding the models, why you did it for only 50 days however, your work was around 100 days.

Response 12: Thank you sir, for the kinetic models and ANN, theoretically, we have to use result from a batch study, which is Biomethane Potential Test. The result can be referred from table 8 (biogas production) and figure 9 (experimental). The basis of this study is the biogas were monitored for 50 days after one-time feed mode of the substrate. The data for the 100 days is a semi-continuous study which cannot be used for machine learning evaluation.

  1. Remove figure 13, you do not need it.

Response 13: Noted Sir, your suggestion is highly appreciated.  We have removed the figure. 

  1. The article needs a thorough review of the English language.

Response 14: Thank you, we have revised the language accordingly.

Reviewer 2 Report

The article entitled “Influence of sugar-rich molasses residue on treatment of cow manure in an Anaerobic Filter reactor with Perforated Weed Membrane and a Conventional Reactor: From start-up to optimum organic loading rate and a machine learning application”

The abstract should be rewritten and better identify the processes with which the experiment has been carried out, indicating the operating conditions, on the other hand it speaks of OLR and it is not clear what it refers to.

At no time is the quality of the effluents from the processes indicated.

Given the number of abbreviations used during the article, it would be interesting to include a table of abbreviations for a correct understanding of it.

The objectives set out in the article should be clearer, including operating conditions and using fewer abbreviations.

There is talk of obtaining optimum performance when perhaps it refers to the best operating conditions, it is not clear to me.

Table 4 is not necessary, it can be commented on in the text

It is difficult to understand the description of how the plant operated, a diagram of operating conditions could be made together with figures 1, 2 and 3, in general it is difficult to understand the experimental methodology that has been followed during the test, it should be clarified.

In Table 6, it should be justified why these three adjustment models have been chosen.

On the other hand, searching in bibliography, the main author of the process has similar articles on the subject, she should indicate in this article what are the scientific novelties of the same from the technological point of view for its acceptance.

Round 2

Reviewer 2 Report

The authors have reviewed the required comments, so the editor could publish the article in its current state